# Effects of Bioactive Dietary Components on Changes in Lipid and Liver Parameters in Women after Bariatric Surgery and Procedures

**DOI:** 10.3390/nu16091379

**Published:** 2024-05-01

**Authors:** Edyta Barbara Balejko, Anna Bogacka, Jarosław Lichota, Jan Pawlus

**Affiliations:** 1Department of Commodity Science, Quality Assessment, Process Engineering and Human Nutrition, West Pomeranian University of Technology in Szczecin, 71-459 Szczecin, Poland; 2Unii Lubelskiej 1, Department of General, Minimally Invasive and Gastroenterological Surgery, Independent Public Clinical Hospital No. 1 of Pomeranian Medical University in Szczecin, 71-252 Szczecin, Poland

**Keywords:** obesity, bariatric surgery, bioactive dietary components, metabolic changes, inflammation, liver dysfunction

## Abstract

Excess adipose tissue, as well as its distribution, correlates strongly with disorders of lipid and liver parameters and chronic inflammation. The pathophysiology of metabolic diseases caused by obesity is associated with the dysfunction of visceral adipose tissue. Effective and alternative interventions such as the Bioenteric Intragastric Balloon and bariatric surgeries such as the Roux-en-Y gastric bypass. The aim of this study was to assess the effect of modifying the recommended standard weight loss diet after bariatric surgery and procedures on reducing chronic inflammation in overweight patients. In the study, bioactive anti-inflammatory dietary components were used supportively. Changes in the concentrations of lipid parameters, liver parameters, antioxidant enzymes, cytokines, and chemokines were demonstrated. The enrichment of the diet, after bariatric surgery, with the addition of n-3 EFAs(Essential Fatty Acids), bioflavonoids, vitamins, and synbiotics resulted in higher weight losses in the patients in the study with a simultaneous reduction in parameters indicating liver dysfunction.

## 1. Introduction

Bariatric surgery is not an ideal form of obesity treatment, but it is often the only one that offers survival and improves the quality of life for patients with morbid obesity. Depending on the BMI (Body Mass Index) value, obese patients are offered bariatric surgery and/or procedures. Among the procedures, the Bioenteric Intragastric Balloon (BIB) has been popular for many years. It is considered the least invasive and reversible and is aimed at patients with grade I obesity (BMI = 30.0–34.9 kg/m^2^), and in special cases as the first stage of weight reduction for patients with grade II obesity (BMI = 35.0–39.9 kg/m^2^) preparing for surgery. Laparoscopically performed RYGB gastric bypass, on the other hand, is a popular method of bariatric surgery, aimed at patients with grades II and III obesity (BMI ≥ 40 kg/m^2^). It involves the simultaneous reduction of the stomach and restriction of absorption by bypassing a significant portion of the intestine. It allows for greater weight reduction, improvement in metabolic parameters including liver and lipids, significant glycemic control, and often a complete remission of type II diabetes [1].

Adipose tissue plays an important metabolic role. Adipocytes are a source of pro-inflammatory biologically active proteins. Excess visceral adipose tissue induces inflammation. In addition, macrophages in obesity-associated inflammation increase the synthesis of pro-inflammatory adipokines. Adipose tissue dysfunction is characterized by decreased release of homeostatic protective factors such as adiponectin, nitric oxide, or protective prostaglandins with the concomitant increased activation of the pathological release of adipokines such as resistin and visfatin and pro-inflammatory interleukins. Locally and systemically, it causes multiple dysfunctions in glucose and lipid metabolism. Chronic inflammation increases the risk of developing or exacerbating symptoms in the course of hypertension, diabetes, atherosclerosis, and metabolic syndrome, including dyslipidemia. Metabolic, hormonal, and oxidative changes after bariatric surgery initially cause elevations in lipid metabolism parameters and liver enzymes. The aim of this study was to assess the effect of a modified recommended standard weight loss diet after bariatric surgery and procedures on reducing chronic inflammation in overweight patients. It was hypothesized that the enrichment of the diet after bariatric surgery with the addition of n-3 EFAs, bioflavonoids, vitamins, and synbiotics could result in long-term comparable, or greater, weight loss in the subjects with a concomitant reduction in parameters indicative of liver dysfunction.

## 2. Materials and Methods

The study included 80 women with a mean age of 51 ± 5.4 years, treated at the Sonomed Medical Center and the Department of General and Vascular Surgery of the University Clinical Hospital at Pomeranian Medical University in Szczecin. The observation time was 12 months. Patients were divided into 4 groups: I—patients after BIB using a standard diet (S), II—patients after BIB using an experimental diet (E), III—patients after RYGB with a standard diet (S), and IV—patients after RYGB with the experimental diet (E). The mean BMI of the subjects was 33.79 ± 5.6 kg/m^2^. All patients had made multiple unsuccessful attempts to lose weight with conservative methods, meaning diet. The standard diet complied with the Polish Society for the Treatment of Obesity’s requirements for feeding people after surgery and bariatric procedures [2]. The experimental diet is the author’s modification of dietary recommendations with increased antioxidant and anti-inflammatory potential [3]. The protein content of the experimental diet (E) was 80–85 g/day, the carbohydrate total was 70 g/day, and the total fat content was 42 g/day. The n-6/n-3 EFA ratio was 1:84. In the experimental diet, shakes, mousses, and purees consisting of products with high antioxidant potential were included. Their antioxidant capacity (TEAC—trolox equivalent antioxidant capacity), ferric reducing antioxidant power (FRAP—ferric reducing antioxidant power), and total polyphenol content from 120.51 mg catechin/100 mL to 250.51 mg catechin/100 mL were determined. [3]. The caloric value of the diets (S, E) in the 1st month was approx. 400 kcal, in the 3rd month it was approx. 600–800 kcal, and in the 12th month it was approx. 1200 kcal. In addition, patients following the experimental diet (groups II and IV) took a synbiotic once a day, every other month during the study, with the following composition: the probiotics *Bifidobacterium lactis; Lactobacillus acidophilus*; *Lactobacillus paracasei*; *Lactobacillus plantarum; Lactobacillus salivarius; and Lactobacillus lactis*; and the prebiotics fructooligosaccharides (FOS) and inulin. The total number of colony-forming units (CFUs) was 1 × 10^9^ CFUs.

Anthropometric measurements were taken using a CEO 123-certified IOI 353 analyzer, which complies with MDD 93/42/EEC medical device directives. The weight and visceral fat area were determined. Most examinations were performed before the start of treatment and diet and at the end of the follow-up period. Patients gave voluntary consent to participate in the study. The study design was approved by the Bioethics Committee of the Regional Chamber of Medicine in Szczecin OIL-Sz/Mf/KB/452/06/05/2015.

The standard panels of liver enzymes aspartate transferase (AST) and alanine transferase (ALT) with total values of CH(Total Chjolesterol), HDL(High-density Lipoproteins), LDL(Low-density Lipoproteines), and TG(Triglycerides) via the colorimetric method were determined in all groups of patients from the collected blood. Assays were performed before the start of treatment and at the 3rd and 12th months. Analyses were performed at accredited diagnostic laboratories.

In erythrocytes, the activity of antioxidant enzymes was evaluated via the spectrophotometric method:-Superoxide dismutase (SOD), using the ability to oxidize adrenaline to adenochrome [4].-Catalase (CAT), assessing the enzyme’s ability to degrade superoxides [5].-Glutathione peroxidase (GSH-Px), which reduces hydrogen peroxide to water with reduced glutathione [6].

Plasma malondialdehyde (MDA) concentration was determined via the spectrophotometric method according to Rice-Evans et al. without chromogen extraction [7].

The immunoenzymatic ELISA technique was used to determine the adiponectin, resolvin, and chemokine contents in serum. The absorbance was read at 450 nm ± 2 nm using a multi-channel ELISA reader, ELx808 (BIO-TEK Instruments, Inc., Winooski, VT, USA). The KC Junior for Windows calculator program from BIO-TEK Instruments, Inc., USA, was used to develop calibration curves and calculate the concentration values of the parameters tested. The following commercial kits were used for the study:-Adiponectin (catalog No.: EO605h, EIAab Science Co., Wuhan, China; method accuracy was ± 1% ± 0.010 absorbance with a range of 0.156–10 ng/mL;-Resolvin D1 (RvD1) series (catalog No.: 500380, Cayman Chemical Co., Ann Arbor, MI, USA); method accuracy was ± 1% +/− 0.010 absorbance with a range of 3.3–2000 pg/mL;-TNF-α (catalog No.: EIA-4641, DRG Diagnostics Int., Inc., Springfield, NJ, USA); method accuracy was ± 1% ± 0.010 absorbance with a measurement range of 7–500 pg/mL;-TGF- β1, (catalog No.: EIA-1864, DRG Diagnostics Int., Inc., Springfield, NJ, USA); method accuracy was ± 1% ± 0.010 absorbance with a measurement range of 1.9–600 pg/mL;-IL-1 β, (catalog No.: EIA-4437, DRG Diagnostics Int., Inc., Springfield, NJ, USA); method accuracy was ± 1% ± 0.010 absorbance, with a measurement range of 0.35–1200 pg/mL;-IL-10, (catalog No.: EIA-4699, DRG Diagnostics Int., Inc., Springfield, NJ, USA); method accuracy was ± 1% ± 0.010 absorbance, with a measurement range of 1.6–2000 pg/mL;-IL-6 (catalog No.: EIA-4640, DRG Diagnostics Int., Inc., Springfield, NJ, USA); the accuracy of the method was ± 1% ± 0.010 of absorbance, with a measurement range of 1.6–2000 pg/mL.

### Statistical Analysis

The statistical analysis of the results of the study was performed using Statistica version 13 software. To show statistically significant differences in concentrations between the groups, the non-parametric Wilcoxon paired rank-order test was used. Statistical significance was set at a level of *p* < 0.05.

## 3. Results

We compared the effects of weight reduction using the bioelectrical impedance method in the four study groups before surgery and after one year of follow-up (Figure 1 and Figure 2).

The BMI value on the day of the BIB procedure in group I patients following a standard diet averaged 31.12 ± 3.7 kg/m^2^, with an average visceral fat area of 163 ± 12.1 cm^2^. After one year of follow-up, the BMI in this group decreased to 27.34 ± 6.02 kg/m^2^, and the visceral fat volume decreased to 135.6 ± 4 cm^2^. The BMI value on the day of BIB surgery in group II patients on the modified diet averaged 32.18 ± 4.9 kg/m^2^, and the visceral fat area averaged 172.5 ± 27.21 cm^2^. After one year of follow-up, the average BMI decreased to 25.12 ± 4.1 kg/m^2^, and the average visceral fat volume decreased to 120.8 ± 15.7 cm^2^. The BMI of patients in group III on the day of RYGB surgery who followed a standard diet averaged 36.7 ± 3.9 kg/m^2^, and the visceral fat area averaged 177.8 ± 17.5 cm^2^. After one year of follow-up, the BMI average decreased to 28.4 ± 3.5 kg/m^2^ and the average visceral fat volume decreased to 127.6 ± 6.2 cm^2^. The BMI of patients on the experimental diet (group IV) on the day of RYGB surgery averaged 35.16 ± 4.8 kg/m^2^, and the visceral fat area averaged 184.2 ± 12.6 cm^2^. After one year of follow-up, the average BMI decreased to 25.9 ± 3.7 kg/m^2^, and the average visceral fat volume decreased to 112 ± 13.7 cm^2^.

Table 1 shows the mean results and standard deviations of the obtained lipid parameters and liver enzymes in the sera of the four groups of patients. In all patients, abnormally elevated levels of all determined lipid fractions and liver enzymes were noted before the procedures and surgeries. After the 3rd and 12th months of balloon insertion into the stomach, slight decreases in the mean concentrations of all parameters studied were observed in the serum of patients following a standard diet (group I). These were statistically insignificant differences. Only the mean TG concentration, which was initially 155 ± 14.3 and 109.14 ± 10.7 after the 3rd month, increased to 115.12 ± 16.2 [mg/dL] after the 12th month. In patients after BIB following the experimental diet (group II), there was a significant statistical increase in HDL cholesterol levels from 39.8 ± 15.2 to 50.16 ± 5.4 [mg/dL]. Other mean concentrations of lipid fractions and liver enzymes decreased significantly. The analysis of the studied parameters showed, in post-RYGB patients following the standard diet (group III), statistically significant differences in the mean concentrations of total cholesterol and TG. In contrast, there were no significant differences in HDL or LDL cholesterol concentrations over the course of the study. ALT and AST enzyme activities increased after 3 months to decrease after one year of follow-up. In patients after RYGB and following the experimental diet (group IV), the mean values of all parameters were statistically significantly different. In this group of subjects, the activity of ALT and AST enzymes gradually decreased during the 12 months of follow-up. There were also statistically significant differences between the results of groups III and IV due to the differences in the diets.

Statistically significant differences were found between groups I(S) and II(E) of total cholesterol, HDL, LDL, and TG fractions. In addition, statistically significant differences were also found in lipid fractions between groups III(S) and IV(E) HDL, LDL, TG, and enzymes ALT and AST.

We evaluated oxidative stress in obese patients by analyzing the activity of antioxidant enzymes and the degree of lipid peroxidation. Oxidative stress results in lipid peroxidation. Peroxidation is the oxidation of unsaturated fatty acids that make up phospholipids, of which MDA is a marker. Antioxidant enzymes are the first line of defense against reactive oxygen species. In our study, before BIB surgery and RYGB surgery, a significant increase in the severity of oxidative stress was observed in all obese patients. Table 2 shows the results of superoxide dismutase (SOD), catalase (CAT), and glutathione peroxidase (GSH-Px) enzyme activities and malondialdehyde (MDA) levels.

In patients after BIB and RYGB, a reduction in serum MDA levels was noted in addition to a reduction in body weight and body fat. Dietary components with antioxidant properties may reduce patients’ exposure to oxidative stress associated with obesity. In our study, the changing activity of antioxidant enzymes was noted (Table 2).

Statistically significant differences were found between groups I(S) and II(E) of SOD, GSH-Px, and MDA. In addition, statistically significant differences were also found between groups III(S) and IV(E) in CAT, SOD, GSH-Px, and MDA.

The results of the mean serum cytokine concentrations of patients in all groups are shown in Table 3. High concentrations of pro-inflammatory cytokines occurred in all patients before therapy. These results correlated positively with excess body weight and body fat before BIB and RYGB surgery. With the remission of obesity, a decrease in pro-inflammatory cytokines and an increase in anti-inflammatory cytokines were observed in both groups. The selection of appropriate bioactive components of the diet contributed to the normalization of inflammatory factors in obese patients.

The statistical analysis of cytokine results showed no statistically significant differences only in TGF-β and IL-1β levels.

Assuming that the diet has anti-inflammatory properties, the serum levels of adiponectin and resolvins were determined in patients (Table 4). An increase in adipokine levels was observed in all obese patients after BIB and RYGB. Statistically significant differences were shown in groups I vs. II and III vs. IV, depending on the diet used. Moreover, it was confirmed that higher adiponectin concentrations correlate with anti-inflammatory IL-10. In addition, resolvins are involved in extinguishing the acute phase of inflammation. The lowest concentrations of resolvins were obtained in all obese patients before BIB and RYGB. In the blood of Group II patients after the observation period, a small, not statistically significant increase in resolvin D concentrations was obtained. Significantly higher concentrations of the tested anti-inflammatory mediator were recorded in group IV after the end of therapy. It is presumed that such an effect was the result of a diet with antioxidant properties containing DHA. Resolvin D has an immunosuppressive effect on the release of pro-inflammatory cytokines [8], so this was also the subject of analysis in our study. The concentrations of both adipokines were inversely proportional to the elevated parameters of existing inflammation.

Statistical analysis of the results of adiponectin and resolvin values in the four groups analyzed showed statistically significant differences between them.

## 4. Discussion

Based on the possibility of using functional ingredients in supporting the treatment of obesity, an attempt was made to formulate a diet with functional characteristics. Bioactive compounds such as flavonoids exhibit broad antioxidant, anti-inflammatory, and hepatoprotective effects. Research indicates that dedicated diets can be used to effectively reduce body fat and inflammation.

It is known that excess body fat, as well as unfavorable fat distribution, induces chronic inflammation [9]. In most studies, authors focus on the results of reducing lipid and/or liver parameters. In our study, we attempted to demonstrate the effect of selected dietary components as a supportive element of bariatric surgery and procedures, on changing inflammatory parameters and hepatic dysfunction. For this purpose, the diet was composed by way of increasing the proportion of bioflavonoids, vitamins, minerals, fatty acids, and probiotics giving it bioactive and immunomodulatory characteristics [3].

Adiposopathy, or adipose tissue dysfunction, is characterized by the production of inflammatory cytokines and chemokines. The cause of organ dysfunction in obesity disease is oxidative stress expressed via the increased synthesis of reactive oxygen species (ROS) [10]. Oxidative stress leads to the development of non-alcoholic steatohepatitis (NASH). Our studies have shown changes in the activity of antioxidant enzymes and lipid peroxidation, indicating a reduction in oxidative stress. The strongest satisfactory effect was obtained after RYGB surgery in patients following the experimental diet. Obese patients have a threefold higher incidence of non-alcoholic steatohepatitis, with a twofold higher complication rate of cirrhosis or bile duct stones with concomitant conditions of severe acute pancreatitis [11,12]. In addition, the long-term adherence to a low-energy diet is a contributing factor to liver fatness.

One of the patient’s qualifying factors for surgery is the requirement for a 10% loss in baseline body weight [13]. Often, patients starve themselves for many months to achieve the necessary reduction. For the next three months after surgery, a feeding regimen that is energy-deficient is in effect. The rate of weight loss after surgery is significantly faster than the physiological rate. Therefore, after operations, increasing liver enzyme activity is observed, as well as an increase in liver fatness on ultrasound imaging. This is due to the action of lipase. Rapid weight loss causes the rapid release of a large number of lipid cells from adipose tissue into the bloodstream, resulting in hepatic steatosis. In addition, the thickening of bile and its backlog results in a 35% higher risk of gallstones [14]. In our study, ALT and AST enzyme activities initially increased in patients after RYGB. Upon longer follow-up, satisfactory low levels of lipid parameters and liver enzymes were achieved. After bariatric surgery combined with dietary care, an improvement in the ratio of lipid fractions and a decrease in liver enzymes is achieved [15,16,17]. The inflammation that accompanies obesity promotes pathological changes in the liver. Dietary care in self-observation had a preventive goal against NASH. In a proper diet, the appropriate ratio of EFA n-6 /n-3 should be 4:1. An excess of n-6 fatty acids in the diet inhibits the metabolism of n-3 fatty acids so that the balance between pro- and anti-inflammatory lipid mediators is disrupted. Giving obese patients increased amounts of n-3 fatty acids in an experimental diet was thought to reduce the inflammatory process. In 2004, Bays defined a new concept, aiming not only to reduce fat mass but also to correct adiposopathy [18]. The high correlation between obesity and the incidence of abnormal metabolic parameters, e.g., lipid profile, blood glucose or liver enzymes, and insulin resistance, is due to the endocrine function of adipose tissue—the secretion of adipokines [19].

Adipocytokines affect all phases of the immune response, regulating the proliferation, differentiation, and activation of B lymphocytes, T cells, NK cells, monocytes/macrophages, and granulocytes. Increased macrophage infiltration and increased levels of the cytokines or chemokines MCP-1, IL-6, IFN-γ, and TNF-α are observed. Buechler et al. described the involvement of TGF-β in liver fibrosis. Blocking TGF-β signaling protects against obesity, insulin resistance, and hepatic steatosis [20]. In our study, TGF-β1 levels were higher in all patients before BIB and RYBG.

Adipocytokines are involved in glucose and lipid metabolism. Adiponectin is synthesized mainly in adipocytes. Two isoforms of membrane receptors for adiponectin AdipoR1 and AdipoR2 are located in the liver. It is involved in glucose and fatty acid metabolism in the liver and muscle. A reduction in acetyl-CoA (ACC) activity in the skeletal muscle and liver by both AdipoR1 and AdipoR2 receptors stimulates and enhances the β-oxidation of fatty acids, with the direct consequence of lowering serum triglyceride levels [21]. It inhibits free radical production, TNF-α, IL-8 reduction, and LDL cholesterol oxidation. An IL-1 receptor antagonist reduces TNF-α release, as confirmed in our study. At the same time, it increases IL-10 levels, thus exhibiting anti-atherosclerotic and anti-inflammatory functions and increasing insulin resistance [22,23]. Adiponectin protects the liver from steatosis and inflammation by directly suppressing TNF-α and decreases the concentration of free fatty acids [24]. In adipocytes, TNF-α inhibits the esterification capacity of fatty acids.

In our study, adiponectin and resolvin levels increased in the blood of post-RYGB patients following an anti-inflammatory diet. D-resolvins play an important role in the metabolic syndrome associated with obesity [25] and show protective effects in inflammation caused by oxidative stress [26]. Titos demonstrated the immunoregulatory effect of D1 resolvins on IL-10 levels in inflamed adipose tissue. RvD1 inhibited uncontrolled adipose tissue inflammation in obese subjects. According to Titos et al., RvD1 can treat obesity-related insulin resistance and other metabolic complications, reducing adipose tissue inflammation by lowering the concentrations of cytokines and adipokines (TNF-α, IL-6, IL-8, IL-1β, MCP-1), and reducing inflammation in obesity-induced NASH [27,28]. In patients of both groups after one year of follow-up, higher levels of resolvin were observed. The highest in patients after RYGB. Statistically significant differences were shown in resolvin D concentrations between groups III and IV patients. The experimental diet in the author’s modification contained 28% PUFA, where the amount of n-3 acids was 9.87% of rations, and n-6 was 18.20% of rations, achieving a favorable ratio (n-6)/(n-3) of PUFA = 1:84. Changing the profile of the diet and enriching it with n-3 fatty acids, such as eicosapentaenoic acid and docosahexaenoic acid increased the synthesis of resolvin D1. The obtained values of resolvin D correlated with a decrease in total TG and LDL cholesterol and an increase in HDL in group IV. In addition, visceral fat reduction and dietary components regulating adiponectin and resolvins had a cascading effect on the regulation of pro- and anti-inflammatory cytokines, thereby minimizing chronic inflammation. It is noteworthy that the deficiency in resolvins in obesity appears to be a generalized defect in all tissues, as deficiencies are found in blood, liver, and skeletal muscle in addition to adipose tissue [29]. The effects of enriching the diet with n-3 fatty acids in patients with associated inflammation and the beneficial effects of resolvins are pointed out by Moro et al. [30]. Resolvin D is considered an immunomodulator in chronic inflammation caused by hypercholesterolemia. The progression of simple steatosis to NASH is associated with the chronic exposure of hepatocytes to pro-inflammatory cytokines. The main inflammatory cytokines during these lesions are TNF-α, IL-6, and IL-1β, and this can result in the development of inflammation and fibrosis [31]. Our study showed a statistically significant decrease in the concentrations of pro-inflammatory cytokines, which can contribute to the apoptosis and necrosis of hepatocytes, neutrophil chemotaxis, the activation of stellate cells, and the production of Mallory bodies [32].

The production and release of pro-inflammatory cytokines, both peripherally and locally in the liver, are associated with several factors. The most relevant are lipotoxicity induced via an increase in free fatty acids, insulin resistance, impaired adipose tissue metabolism, and the presence of toxins released from the gut [33]. Numerous studies have shown a correlation between dysbiosis and changes in lipid profiles in obese individuals. Li et al. observed the high efficacy of an applied probiotic supplement in obesity. He showed a decrease in triglycerides and total cholesterol, and an inhibition of serum ALAT and ASPAT activity in probiotic users. Dietary supplementation with *L. acidophilus*, *B. bifidum* BGN4, and *B. longum* BORI reduced hepatocyte hydrops degeneration and decreased hepatic steatosis [34]. Dysbiosis can impair lipid metabolism. It has been shown that the physiological bacterial flora participates in the digestion of dietary macronutrients, absorption of micronutrients, and fermentation of dietary ballast components. In addition, it lowers the concentration of triglycerides; participates in the hydrolysis of cholesterol through supplied enzymes; and prevents obesity, fatty liver, and type II diabetes [35,36]. The beneficial effect of *Lactobacillus* and *Bifidobacterium* lies in the improvement in body function resulting from the reduction in one or more risk factors for disease development. Supplementation with the *Lactobacillus casei* strain *Shirota*, compared to Orlistat, resulted in a greater reduction in fat mass; in addition, it reduced ALAT levels, a marker of liver cell damage [37]. Probiotics in our study had an immunomodulatory effect, stimulating the production of anti-inflammatory cytokines. They are involved in fat mass reduction. In our study, the experimental diet was used, containing probiotics and appropriate proportions of fatty acids, having antioxidant and anti-inflammatory properties, and regulating parameters of lipid metabolism, liver enzymes, and inflammation. The result of the appropriate management of patients during the weight loss process was a satisfactory weight reduction resulting from an adaptation to the new dietary regime. Significantly better results were obtained in patients with significant visceral fat loss after RYGB surgery and following the experimental diet.

In the study by Lu et al., the effects of selected bioactive dietary components on obesity were analyzed and the molecular mechanisms were assessed in cellular, animal, and human models. Bioactive compounds from spices such as cinnamon, rosemary, ginger, saffron, and turmeric assisted in reducing body weight. In addition, they reduced lipid accumulation in adipose cells by regulating the expression of transcription factors such as CCAAT-binding protein/enhancer of gene expression for granulocytes (C/EBP) and peroxisome proliferator-activated receptor (PPARγ). They may also modulate the activities of certain lipogenesis-related enzymes, such as acyl-CoA carboxylase (ACC), fatty acid synthase (FAS), glycerol-3-phosphate dehydrogenase (GPDH), and others. After the oral administration of spice extracts, increased thermogenesis in adipose tissue has been demonstrated, with a concomitant reduction in human body weight [38,39]. Recent results have shown the beneficial actions of the inverse relationship between anthocyanin and flavonoid intake, and the risk of obesity and diabetes has been proposed [39]. Many studies confirm the potential of plant-based bioactive dietary ingredients to reduce adipogenesis and lipid accumulation by interfering with several enzymes and markers involved in lipid formation and breakdown in adipocytes [40].

The study evaluated the effects of a modified weight loss standard diet. Numerous studies worldwide have confirmed the beneficial effects of functional food ingredients on weight loss. Currently, new diets are being sought that produce long-term fat reduction effects. Knowing the pathomechanism of obesity, it is important to consider not only weight reduction but also the reduction in inflammation.

## 5. Conclusions

The study showed a correlation of lipid, hepatic and inflammatory disorders in patients with excess body fat. 

With the reduction of visceral adipose tissue, inflammation decreased. Changes in the concentrations of lipid parameters, liver parameters, antioxidant enzymes , cytokines and chemokines were shown. After enrichment of the diet with n-3 EFAs and antioxidants, there was greater weight loss in the patients studied, with a more effective reduction in parameters indicative of liver dysfunction.

## Figures and Tables

**Figure 1 nutrients-16-01379-f001:**
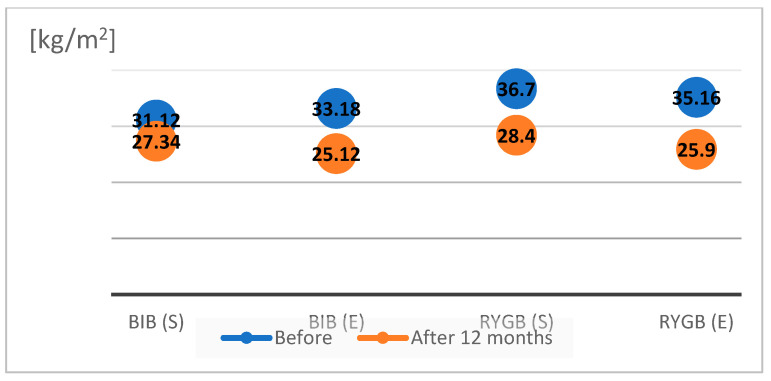
BMI values of patients before and after BIB and RYGB.

**Figure 2 nutrients-16-01379-f002:**
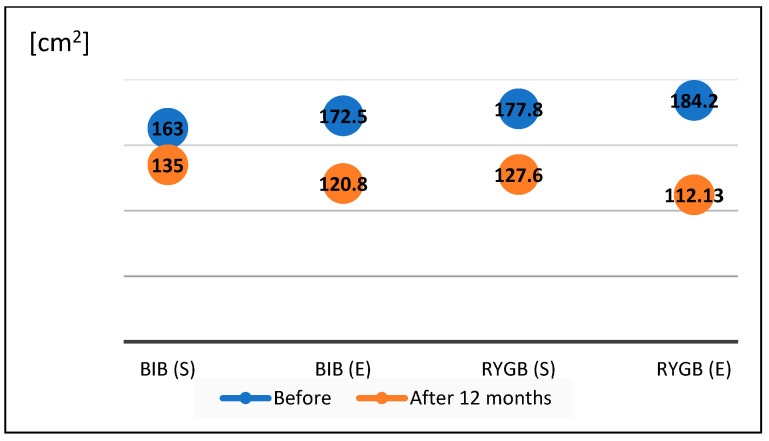
Visceral fat area of patients before and after BIB and RYGB.

**Table 1 nutrients-16-01379-t001:** Serum levels of lipid fractions and liver enzymes in patients, before and after the BIB and RYGB after 12 months.

Parameters Measured	BIB (S)	BIB (E)	*p* Value	Statistical Significance	RYGB (S)	RYGB (E)	*p* Value	Statistical Significance
	$\bar{x}$ ± SD	$\bar{x}$ ± SD			$\bar{x}$ ± SD	$\bar{x}$ ± SD		
TC [mg/dL]	190.8 ± 12.11	154 ± 3.6	0.005	S	159.8 ± 27.3	152.5 ± 12.1	0.005	S
HDL [mg/dL]	38. 7 ± 5.12	50.16 ± 5.4	0.0001	S	40.2 ± 12.6	52.5 ± 12.1	0.016	S
LDL [mg/dL]	147.2 ± 11.3	108.12 ± 11.3	0.043	S	110.4 ± 11.7	90.4 ± 7.8	0.000132	S
TG [mg/dL]	115.12 ± 16. 2	88.17 ± 10.7	0.000089	S	99.30 ± 23.7	62.12 ± 13.7	0.004	S
ALT [U/L]	28.7 ± 7.6	27.5 ± 9.7	0.806	NS	29.8 ± 7.5	22.12 ± 13.7	0.000103	S
AST [U/L]	24.16 ± 5.3	22.7 ± 6.2	0.99	NS	20.5 ± 6.4	17.54 ± 8.7	0.0112	S

**Table 2 nutrients-16-01379-t002:** Antioxidant enzyme activity in patients before and after BIB and RYGB.

Parameters Measured	BIB (S)	BIB (E)	*p* Value	Statistical Significance	RYGB (S)	RYGB (E)	*p* Value	Statistical Significance
	$\bar{x}$ ± SD	$\bar{x}$ ± SD			$\bar{x}$ ± SD	$\bar{x}$ ± SD		
CAT [A/g Hb]	183 ± 15	183 ± 21	0.806	NS	199 ± 25	316 ± 12	0.00036	S
SOD [U/g Hb]	1213.93 ± 45.7	1778.76 ± 15.3	0.0034	S	1414.53 ± 38	2089.64 ± 21.4	0.0017	S
GSH-Px [U/g Hb]	7.58 ± 1.5	10.39 ± 3.3	0.005	S	7.26 ± 1.3	12.6 ± 4.9	0.0348	S
MDA [µmol/L]	7.37 ± 1.2	5.10 ± 1.3	0.0026	S	6.25 ± 1.7	3.49 ± 2.1	0.0366	S

**Table 3 nutrients-16-01379-t003:** Serum cytokine levels in patients, in the four groups before and after BIB surgery and RYGB surgery and the diets used.

Parameters Measured	BIB (S)	BIB (E)	*p* Value	Statistical Significance	RYGB (S)	RYGB (E)	*p* Value	Statistical Significance
	$\bar{x}$ ± SD	$\bar{x}$ ± SD			$\bar{x}$ ± SD	$\bar{x}$ ± SD		
TNF-α	13.23 ± 1.7	8.8 ± 4.5	0.005	S	15.23 ± 3.7	7.01 ± 5.3	0.0034	S
TGF-β	3.6 ± 0.9	3.2 ± 1.1	0.82	NS	4.5 ± 1.2	3.1 ± 1	0.01	S
IL-1β	36.6 ± 2.5	32.1 ± 3.3	0.96	NS	35.6 ± 6	24.13 ± 5.3	0.036	S
IL-10	6.2 ± 3	10.42 ± 1.3	0.005	S	7.5 ± 2	15.3 ± 2.1	0.0011	S
IL-6	18.41 ± 3.6	12.5 ± 3.1	0.0005	S	18.4 ± 2.2	10.31 ± 1.2	0.012	S

**Table 4 nutrients-16-01379-t004:** Mean values of serum levels of adiponectin [ng/mL] and resolvins [pg/mL], in patients in the four groups before and after BIB and RYGB and the diets used.

Parameters Measured	BIB (S)	BIB (E)	*p* Value	Statistical Significance	RYGB (S)	RYGB (E)	*p* Value	Statistical Significance
	$\bar{x}$ ± SD	$\bar{x}$ ± SD			$\bar{x}$ ± SD	$\bar{x}$ ± SD		
Adiponectin	10.43 ± 3.52	16.88 ± 2.5	0.0017	S	12.2 ± 3.6	21.2 ± 4.5	0.000029	S
[ng/mL]								
Resolvins	35.56 ± 3	45.55 ± 5	0.00005	S	37.46 ± 5	47.34 ± 5.1	0.000145	S
[pg/mL]								

## Data Availability

Data are contained within the article.

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
