# Peer review of "Effects of Bioactive Dietary Components on Changes in Lipid and Liver Parameters in Women after Bariatric Surgery and Procedures"

_nutrients, 2024, doi:10.3390/nu16091379_

Round 1
Reviewer 1 Report
Comments and Suggestions for Authors
Overall, due to the presentation of the study, it is difficult to determine what the purpose, methods, results are and the overall outcomes. Some notes are listed below.
Introduction:
-provide table for grading of obesity - how it's determined
-Include purpose and hypothesis for the study in the introduction.
-provide background info on why interventions are being tested - antioxidants, etc.
Methods:
-state what Group 1 and Group 2 are
- n6:n3 ratio is 1:84? what is the evidence for this?
-rather than "author's diet" perhaps use "test diet" or "investigational diet"
- what was total kcal of test diet? show table comparing the Standard of care versus test diet for macronutrients, micronutrients, etc.
- spell out acronyms first time appearing. What is TEAC stand for? FRAP?
-what was test diet comprised of - what foods - and then how was antioxidant capacity calculated and determined? is this a validated method?
-why were they taking a synbiotic? how many CFU of bacteria and gm prebiotic were in it?
- show data for "patients found to be dyslipidemic"
- how were meals provided to patients?
- how was compliance with consuming the meals measured?
-statistical analysis - seems should use ANOVA not t-tests
-describe the study design prior listing what group 1 and 2 are
-Table 1: data presentation is difficult to see differences, suggest tables with before/after for each group side by side to easily see comparisons and add the statistical test value as a column next to it
-figures 1-3 do not add anything meaningful and it's difficult to determine what they represent
Comments on the Quality of English LanguageOverall, due to the presentation of the study, it is difficult to determine what the purpose, methods, results are and the overall outcomes.
Author Response
-provide table for grading of obesity - how it's determined
Answer: BMI values added in the introduction
-Include purpose and hypothesis for the study in the introduction.
Answer: Included in accordance with the Reviewer's suggestion
-provide background info on why interventions are being tested - antioxidants, etc.
Answer This information was included in the discussion, more in my own work cited below
Balejko E., Balejko J., Plust D. 2018. Assessment of the effect of dietary modifications and Bioenteric Intragastric Balloon treatment on the changes of some morphological and biochemical Parameters in obese patients. Annals of Nutrition &Metabolism. 73: 290-301
Methods:
-state what Group 1 and Group 2 are
Answer: . The groups are described in the Materials and Methods section
- n6:n3 ratio is 1:84? what is the evidence for this? and
-what was test diet comprised of - what foods - and then how was antioxidant capacity calculated and determined? is this a validated method?
Answer: The detailed composition and nutritional value of the research food was published in the work mantioned above. By not repeating the content recalled.
Balejko E., Balejko J., Plust D. 2018. Assessment of the effect of dietary modifications and Bioenteric Intragastric Balloon treatment on the changes of some morphological and biochemical Parameters in obese patients. Annals of Nutrition &Metabolism. 73: 290-301
Below is some data that I do not want to repeat.
Based on the analyses, the composition of fatty acids was determined in diets of patients from group III. A favourable, pro-healthy composition of fatty acids and appropriate proportions of n-6/n-3 fatty acids were observed: SFA 26,95%, MUFA 44,98 %, n-3 PUFA 9,87%, n-6 PUFA 18,20, (n-6)/(n-3) PUFA 1,84, Fat content 8,4%.
Protein content in authors’ diets was 80-85g/day, carbohydrates 70g/day and total fat 42g/day. Fat content in daily food ratio was influenced by food produce and linseed oil used as a supplement (20ml/day).
To assess the quality of fat in the diet the total oxidation value (Totox) of fat was determined (Tab. 1). Totox conventionally is used to describe total oxidation of oils and fats, characterizing primary decomposition products as peroxide value PV (mg/100g) together with anisidine value AV, which refers to the presence of secondary oxidation products. Totox is calculated by the formula: Totox= 0.26 x PV + AV. The values obtained indicate very high quality of lipids in patients’ diet.
Table 1. Indicators of fat quality in patients’ diet
pH |
PV mgO/100g fat |
AV |
Totox |
|||
SD |
SD |
SD |
1,1922 |
|||
5,4 |
0 |
0,86 |
0,1 |
0,9668 |
0,1 |
Table 2. Antioxidative capacity (TEAC), reducing power (FRAP) and total polyphenols Content in cocktails and meals
sample |
TEAC |
FRAP |
Total polyphenols |
|||
mean [µM TE/100g or 100 ml] |
SD |
mean [µM TE/100g or 100 ml] |
SD |
mean [mg catechin/ 100g or 100 ml] |
SD |
|
1 |
480,02a |
13,12 |
195,23b |
10,12 |
120,51b |
3,47 |
2 |
969,21c |
58,21 |
737,41d |
66,43 |
250,51d |
18,97 |
3 |
611,13b |
24,34 |
452,21c |
44,21 |
159,49c |
2,22 |
4 |
433,22a |
24,18 |
78,50a |
2,23 |
80,00a |
3,35 |
Data in rows marked with the same letter do not differ significantly, Tukey's test, p<0.05
Benzie, I. F. F., Strain, J. J. 1996. The ferric reducing ability of plasma FRAP as a measure of antioxidant power: The FRAP assay. Anal. Bioch. 239: 70–76.
-rather than "author's diet" perhaps use "test diet" or "investigational diet"
Answer: Changed as suggested
- what was total kcal of test diet? show table comparing the Standard of care versus test diet for macronutrients, micronutrients, etc.
Answer: Kcal values added in the Materials and Methods section
- spell out acronyms first time appearing. What is TEAC stand for? FRAP?
Answer: The extension is included in Materials and Methods
-why were they taking a synbiotic? how many CFU of bacteria and gm prebiotic were in it?
Answer: Included in the Materials and Methods section
- how were meals provided to patients?
- how was compliance with consuming the meals measured?
Answer: Patients in the research group followed the recommended diets prescribed by a bariatric dietitian. Visits were frequent and a nutritional interview was used.
-statistical analysis - seems should use ANOVA not t-tests
Answer The study compared two groups of data, so it was decided to use the non-parametric Wilcoxon pairwise order test
-Table 1: data presentation is difficult to see differences, suggest tables with before/after for each group side by side to easily see comparisons and add the statistical test value as a column next to it
-figures 1-3 do not add anything meaningful and it's difficult to determine what they represent

Reviewer 2 Report
Comments and Suggestions for Authors
The authors are describing the effects of bioactive dietary components on changes in lipid and 2 hepatic parameters with associated inflammation in patients after bariatric surgery and procedures. Some informations are missing to make easier to read the manuscript. Please find below the comments.
1 Introduction
More informations must be given about inflammation, obesity and MS. the authors only focused on surgery but they have to justify why there could have a relation between lipid and hepatic paramters and bariatric surgery.
2 Results
Please take care about writing the units (exponents for example).
In Tables 1 to 4, no statistical analysis appears. Please indicate the difference between the means and the p value when different.
Figures 1, 2 and 3: please another presentation (table for example) to have the values of each parameters. We need the activity value to compare the results. Moreover, what is the expression of the units ?
Lines 321 to 326 must be filled.
Author Response
More information on inflammation, obesity and MS should be provided The authors focused only on surgery, but they need to justify why there may be a relationship between lipid and liver parameters and bariatric surgery.
Ans. Information has been added in the introduction. 2 Results.
Please note the notation of units (for example, exponents). ????? No statistical analysis appears in Tables 1 through 4. Please indicate the difference between the means and the p-value if they are different.
Ans. Changed according to the reviewer's suggestion Figures 1, 2 and 3: please have a different presentation (e.g., a table) to have the values of each parameter. We need activity values to compare the results.
Moreover, what is the expression of the units?
Ans. The results are included in a table Please fill in lines 321 to 326.Author Contributions :
Conceptualization, Balejko E.;
Methodology, Balejko E., Bogacka A.;
Validation, Balejko E., Bogacka A.;
Formal Analysis, Balejko E.
Investigation, Balejko E.
Resources, Lichota J., Pawlus J.;
Data Curation, Balejko E.
Writing – Original Draft Preparation, Balejko E.
Writing – Review & Editing, Balejko E.
Supervision, Balejko E.
Project Administration, Translation Lichota J., Bogacka A.
Funding: This research received no external funding
The study was conducted according to the guidelines of the Declaration of Helsinki, and approved by the Institutional Review Board (or Ethics Committee) of Regional Chamber of Physicians in Szczecin on December 10, 2015, is OIL-Sz/MF/KB/452/06/05/2015
The authors declare no conflict of interest.

Reviewer 3 Report
Comments and Suggestions for Authors
The following are my suggestions to improve the publishability of these manuscript.
1. The title sounds confusing to me, especially the "hepatic parameters with associated inflammation". Do the authors mean "hepatic parameters associated with inflammation? Since the study population are women, I also suggest indicating this in the title.
2. The abstract needs some work. Please add more details on the methods, results and conclusion.
3. The introduction needs improvement. I suggest that the authors write about the limitations of bariatric surgery and the need for adjunct/supportive therapy to improve surgical outcomes. I suggest to also provide aims or hypotheses. Also, provide proper citation/s when needed.
4. Can the authors provide the inclusion and exclusion criteria used for this study? Why were only women included in this study?
5. This paper heavily relies on ELISA and other plate-based assay. Can the authors provide performance and validation parameters for the assays used?
6. What are the reasons why the data was analyzed using non-parametric tests? Were the data non-normal in distribution? Also, what are the reasons why the data were split into BIB and RYGB subgroups and not analyzed as a whole dataset? Also, if the authors are analyzing using non-parametric tests, data should be expressed as medians with either percentiles or ranges.
7. I advise the authors to have the changes in BMI, body composition be in a table or graph to improve readability. Also, please include the month 3 data on BMI and body composition. Were there changes in the fat-free mass?
8. In the tables, please indicate statistical significance.
9. Fig 1-3 demonstrating the changes in the markers of oxidative stress look very unconventional. I suggest that the authors show the data, either in a table or bar graphs. Right now, it's hard to evaluate the validity of these data.
10. Does the new diet affect overall calorie intake of the patients? Any changes in hunger, or satiety indices, or any measure of overall acceptance of the new diet.
11. I suggest rewriting the discussion to only focus on discussing the results. A big portion of the discussion is a review of literature which can be moved to the introduction. Also, there are parts of the discussion about NASH. However, the research does not include data about NASH parameters beyond ALT and AST. I also advice to break the last paragraph of the discussion because it is very long.
12. When the authors mentioned that BIB patients returned to pre-surgical body weight, when did this happen? Was there are difference in the rate weight recidivism between normal diet vs new diet?
13. Bariatric surgery often results to improved diabetic status. Was there a difference between normal and new diet?
14. Also indicate that the study was only done in women and that research should be done in a cohort of men in the future.
Minor:
1. Add the word stage after the obesity in Line 29.
2. Please re-write the sentence describing the diet composition in Line 48-49
3. Please follow the journals citation formats
4. The acronym ELISA already means it is an immunoenzymatic assay.
5. Please check the journal rules on decimal points - whether comma is acceptable.
6. Can the authors fill this sections? Author Contributions, Funding, Institutional Review Board Statement, Informed Consent Statement, Data Availability Statement, Conflicts of Interest.
7. Please go over the manuscript several times and comb for grammatical errors. There are numerous.
Author Response
Ad 1. Title changed as suggested by reviewer.
Accompanying inflammation (removed) was related to obesity.
Ad 2. The summary has been revised and supplemented
Ad 3. The introduction has been revised and supplemented
Ad 4. Women have a different body composition than men. Fat and muscle measurements were also taken in the analyses. In this way, acceptable differences based on gender were avoided. Significantly (90%) more women undergo surgery. Women cooperated during follow-up.
Ad 5. The answer has been added in the methodology
Ad 6. The first step in any statistical analysis is to test the normality of the distribution. If the analysis using the normality test shows its absence then the consequence is the use of a non-parametric test this was the case here. Since two study groups were compared, the Wilcoxon paired comparisons test, widely used in medicine, was therefore applied.
BIB is a procedure and RYGB is surgery. They are completely different interventions in the body. The metabolic changes are different after BIB and different after RYGB.
Ad 7. The reviewer's suggestion was followed
Ad 8. Zastosowano siÄ™ do sugestii recenzenta
Ad 9. Figures 1 - 3 turned into tables
Ad 10. The research diet does not differ in caloric content from the standard diet. The selection of products was individual for acceptance, easily digestible. At the same time rich in antioxidants.
The paper cited in the text contains the basic data of the diet. I did not want to repeat them.
Ad 11. Added clarification in the discussion.
Ad 12. Both patients on the research diet and the standard diet gradually gained weight just a few months after the balloon was removed.
Ad 13. Statistically significant changes in glycaemia were observed. The results will be published in a subsequent paper.
Ad 14. Statistically significant changes in glycaemia were observed. The results will be published in a subsequent paper.
Comments submitted as minor were also followed

Round 2
Reviewer 2 Report
Comments and Suggestions for Authors
The authors tried to answer to the recommendations but there are still some information missing:
- in the introduction section, mors informations should be provided concerning metabolic syndrome, obesity and inflammation. What has been added in the last revised version is not enough.
- table 1, there is still error concerning the statistical analysis: please explain how the cholesterol level for the RYGB (S) and RYGB (E) cann have a p value of 0.005 with similar mean. I think that statistical analysis must be confirmed for all the results.
Author Response
Thank you for your valuable comments to which we respond below:
- Information on metabolic syndrome inflammation and obesity in the Introduction section has been revised
- Statistical calculations were reanalyzed. The results are correct.
- Enclosed we send the article after another revision. We will send the final version of the article after language correction at a later date.

Reviewer 3 Report
Comments and Suggestions for Authors
1. About the ELISA performance parameters, are these numbers calculated from the experiments or theoretical according to the manufacturer's validation. I would like to see, co-efficient of variations, if samples were run in replicates and if multiple plates were used.
2. The new figures 1 and 2 is confusing. Instead of a line connecting all treatment groups at a given time point, a line should be connecting the before and after within a treatment group.
3. Significant amount of data was removed in the current version compared to the initial version. In Table 1, the authors now only report the 12 month values - why were the baseline and 3 month observation removed? Similarly, the same thing happened with the cytokines and adipokines.
4. In the antioxidant enzyme activity (Table 2). It seems there is only 1 set of values. Are these before surgery or after surgery? probably the same issue with my comment at #3?
5. The discussion still needs more work. The ideas are incoherent and seem to jump from one subject to another. In addition, although there is direct evidence that bariatric surgery can help alleviate the clinical and biochemical symptoms of NASH, I advise the authors not to focus too much of NASH as only ALT and AST were measured. No data were presented on liver fat. Focus on the roles of the dietary components and how these benefits in inflammatory markers led to health improvements in the patients, especially body weight.
Comments on the Quality of English Language6. Please consult a native English writer. There are multiple grammatical and syntax improvements needed in the manuscript. Also, some words are left not translated to English, e.g. Parametry mierzone.
Author Response
- ELISA measurements were performed in duplicate. Single, separate plates were used. Precision within the series was below 20% CV. According to the vendor's guidelines.
- The graphical interpretation of the data in Figures 1 and 2 has been changed
- and 4. The tables of cytokines, adipokines and enzymes show the final results. As suggested by one of the Reviewers. 5. The role of dietary components on inflammation is included in the discussion. Enclosed we send the article after another revision. We will send the final version of the article after language correction at a later date.
